# A GeV-TeV particle component and the barrier of cosmic-ray sea in the Central Molecular Zone

Xiaoyuan Huang [1,2 ✉], Qiang Yuan[1,2 ✉] & Yi-Zhong Fan [1,2 ✉]

Cosmic rays are important probe of a number of fundamental physical problems such as the acceleration of high and very high energy particles in extreme astrophysical environments. The Galactic center is widely anticipated to be an important cosmic-ray source and the observations of some Imaging Atmospheric Cherenkov Telescopes did successfully reveal a component of TeV-PeV cosmic rays in the vicinity of the Galactic center. Here we report the identification of GeV-TeV cosmic rays in the central molecular zone with the $\gamma$-ray observations of the Fermi Large Area Telescope, whose spectrum and spatial gradient are consistent with that measured by the Imaging Atmospheric Cherenkov Telescopes but the corresponding cosmic-ray energy density is substantially lower than the so-called cosmic-ray sea component, suggesting the presence of a high energy particle accelerator at the Galactic center and the existence of a barrier that can effectively suppress the penetration of the particles from the cosmic-ray sea to the central molecular zone.

[1] Key Laboratory of Dark Matter and Space Astronomy, Purple Mountain Observatory, Chinese Academy of Sciences, Nanjing 210023, China. [2] School of Astronomy and Space Science, University of Science and Technology of China, Hefei, Anhui 230026, China. ✉email: xyhuang@pmo.ac.cn; yuanq@pmo.ac.cn; yzfan@pmo.ac.cn

It is believed that in the Milky Way the cosmic rays (CRs) could be accelerated by shock waves in supernova remnants (e.g.,[1]) or stellar winds of massive stars[2]. Those charged relativistic particles would then propagate diffusively in the Galactic magnetic field, possibly experiencing re-acceleration, convection, spallation, and energy loss processes[3,4]. Such processes would lead to a large-scale, quasi-steady-state CR sea, which distributes relatively smoothly in the Galaxy, as supported by the Fermi Large Area Telescope (Fermi-LAT[5]) observations[6–8].

However, in the proximity of a recent or currently active accelerator, the smoothly distributed CR sea would be overlaid with a component of fresh CRs. Observations of such a fresh CR component will be very important in studying the acceleration, injection, and transportation processes of CRs. The Galactic Center (GC) region contains a supermassive black hole, Sagittarius A⋆, and other types of particle accelerators such as pulsar wind nebulae and supernova remnants. The large-scale bubbles in γ-rays[9], radio[10], and X-rays[11], and the so-called X-ray chimney[12], may be consequences of the energetic activities of Sagittarius A⋆ in the past. The GC region was then widely regarded as a very attractive astrophysical laboratory for studying the cosmic ray astrophysics.

A bright γ-ray point source at the GC, observed by some instruments from GeV to TeV (e.g.,[13]), may indicate an episodic injection of high-energy particles from past activities of the central black hole[14,15]. With the Fermi-LAT data, a nearly symmetric and extended excess component on top of the diffuse background (the so-called Galactic Center GeV excess; GCE[16–18]) was identified, which could be from either dark matter annihilation or a group of unresolved faint point sources like millisecond pulsars[19]. In the central molecular zone (CMZ) region, the observations of High Energy Stereoscopic System (H.E.S.S.[20]) and other Imaging Atmospheric Cherenkov Telescopes (IACTs) reveal a large amount of very-high-energy (VHE) CRs with a hard spectrum and a high density (a factor of ~10 times higher than that measured locally at the Earth), implying an injection of CRs by a source close to the GC[21–23]. Further studies[24–26] show that there is a gradient of the CR density profile, $\propto r^{-\alpha}$ ($\alpha \sim 1$–1.2), via a deconvolution of the γ-ray profile with the gas density in the CMZ. Likely, Sagittarius A⋆ was more active in the past and had accelerated CRs up to PeV energies which diffuse outwards and collide with molecular gas to produce energetic γ-rays. Finally, a component of γ-rays from interactions between the CR sea and the materials in the GC region is expected to present, which is, however, not properly addressed in some analyses.

To reveal the nature of the component discovered by VHE observations, it would be essential to scrutinize different emission components in the CMZ region and to identify possible counterpart in the low-energy γ-ray band[27,28]. Adopting the radio or VHE γ-ray emission from the Galactic ridge as templates, ref. [27] analyzed Fermi-LAT data and reported an emission component that was suggested to come from a population of nonthermal electrons in the Galactic ridge. Reference [28] analyzed the data in the CMZ region, and concluded that a large fraction of the γ-ray emission measured by H.E.S.S. and Fermi-LAT might originate from the interaction of the diffuse Galactic CR sea with the massive molecular clouds in the CMZ, though the possibility of an additional component cannot be ruled out. In such an analysis, the GCE component, which may play an important role in shaping the result, was not included. To critically address the CR population in the most central region, it is necessary to carry out a dedicated morphological and spectral analysis of the Fermi-LAT data in the GC region, with proper incorporation of the GCE component. This is the main goal of this work.

Here, we report the re-analysis of the Fermi-LAT data (see methods, subsection The Fermi-LAT data and subsection Point

sources and the diffuse model components) in the CMZ region and the identification of a component of GeV-TeV CRs, which is likely the low energy part of the TeV-PeV CR component discovered by the IACTs[21–23]. This supports the presence of a high-energy particle accelerator at the GC. We further show that the inferred energy density of CRs in the CMZ region is clearly lower than that from an extrapolation of the CR sea distribution. A natural explanation is that there is a barrier surrounding the CMZ, maybe due to the strong magnetic field in such a region, that can effectively suppress the penetration of particles from outside to the CMZ.

## Results

**Reduced GeV-TeV CR density within the CMZ.** We single out the CMZ region[29] for studying the properties of CRs near the GC, and the rest is referred to as the off-CMZ region (see methods, subsection The CR densities in the CMZ and off-CMZ regions). The red line and associated bands in Fig. 1 show the best-fit and 1σ, 2σ confidence regions of γ-ray fluxes from the neutral pion decay component in the CMZ region using the template calculated by the GALPROP code[30,31]. Similar results have been obtained if we change the GALPROP template to the Planck dust opacity template[32], which can avoid the uncertainty from the $X_{CO}$ factor when converting the gas tracer emissivity to the molecular gas density, as shown by the green line and associated bands in Fig. 1. The spectral index of the CMZ region is harder by $0.19 \pm 0.07$ than that (about 2.68) predicted in the GALPROP Galactic diffuse emission (GDE) model A[33] (see "Methods", subsection Point sources and the diffuse model components), and the integrated energy flux is $(7.05 \pm 0.44) \times 10^{-5}$ MeV cm$^{-2}$ s$^{-1}$ from 8 to 500 GeV. As a comparison, we show in Fig. 1 with dotted lines the predicted

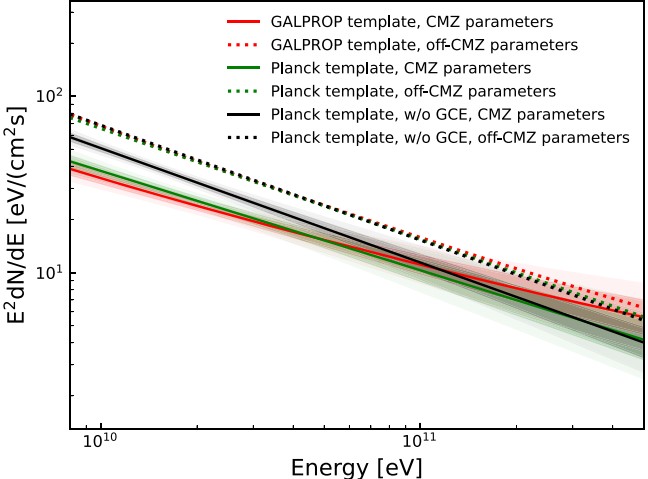

**Fig. 1 Energy flux of γ-ray from neutral pion decays in the CMZ region.** Colored regions with solid lines (red and green) show the fitted results assuming different spatial templates, i.e., the neutral pion decay template from GALPROP and Planck dust opacity template in the CMZ region. Dark and light regions are the 68 and 95% confidence ranges, respectively. Dotted lines show the predicted results using the off-CMZ parameters for given templates, which represent the anticipated CR sea component. The spectrum derived from the CMZ region is harder and the integrated flux is lower than those anticipated from the CR sea interaction. As a comparison, the black line and shaded band show the fitting results derived with the Planck template, but without subtracting the GCE component. The dotted black line shows the anticipated result using the off-CMZ parameters in this case. Again we find that the integrated flux in the CMZ is lower than that anticipated from the CR sea interaction. Source data are provided as a Source Data file.

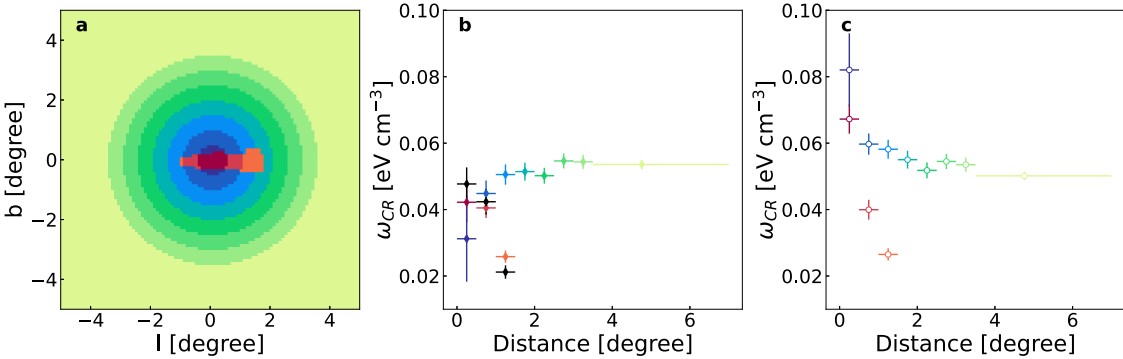

**Fig. 2 The segment division of the GC region and the inferred CR densities distribution. a** The segments, annuli centered on the GC with a width of 0.5° except for the last one, where CR densities are derived, in the CMZ and off-CMZ are marked in red to orange and blue to green, respectively. The same color code is used to show data points of CR densities in corresponding segments in (**b**, **c**). **b** CR densities distribution from fittings with the GCE. For segments outside the CMZ, the CR density is almost a constant. In the CMZ, the CR density declines quickly as the distance increases from the GC. These facts strongly suggest different physical origins of CRs within and outside the CMZ. CR densities in the CMZ, inferred from analysis with the CS map[29], are shown with black points, which agree well with those derived from the fitting with the Planck map. The error bars represent the 1σ statistical uncertainties. **c** CR densities distribution from fittings without the GCE, for which the inferred CR densities near the GC were boosted compared with those shown in (**b**). The error bars represent the 1σ statistical uncertainties. Source data are provided as a Source Data file.

spectrum in the CMZ with the parameters for the off-CMZ region, which is the "expected" emission from interactions between the CR sea and the gas in the CMZ assuming a relatively flat distribution of the sea. The spectral index of the CMZ is clearly harder than that expected from the interaction of CR sea with the gas but is consistent with the VHE $\gamma$-ray observations ($2.32 \pm 0.05_{stat} \pm 0.10_{syst}$ and $2.28 \pm 0.03_{stat} \pm 0.20_{syst}$[24–26]). Furthermore, the integrated flux is lower than that predicted from the CR sea interaction. Our results thus suggest that there is a hard component in the CMZ that possibly coincides with the VHE component, and the CR density of the background sea component in the CMZ region should have been suppressed.

Then we investigate the CR density distribution in the CMZ and off-CMZ directions. To enable a high enough spatial resolution of the gas distribution, the Planck dust opacity map is used as a tracer. We split the Planck map into small segments, as shown in panel (a) of Fig. 2, with a width of 0.5° except for the last one. We fit the normalizations of $\gamma$-ray fluxes for each segment, and derive the CR energy density $\omega_{CR} = w_{CR}(80\,\text{GeV} \le E \le 5\,\text{TeV})$ (see "Methods", subsection The CR densities in the CMZ and off-CMZ regions), as shown in panel (b) of Fig. 2. It is very interesting to find that, for segments outside the CMZ, the CR density is almost a constant, as anticipated in the CR sea scenario. The CR density in the CMZ, however, declines clearly with the increase of distance to the GC, resembling the gradient profile observed at the VHE band. The CR density in the CMZ is also generally lower than that outside the CMZ, confirming the results shown in Fig. 1. The spatial distribution of the CR densities is different from the expectation that the smoothly distributed CR sea would be overlaid with a component of fresh CRs. This may indicate that the contribution of the CR sea has been suppressed in the CMZ, and the emission inside the CMZ mainly comes from the GeV-TeV particles associating with the VHE CR component.

**The GeV-TeV counterpart of the VHE CR component**. The VHE observations of the hard component in the GC show a power-law radial profile, $\propto r^{-\alpha}$, of the CR density with $\alpha = 1.10 \pm 0.12$[24] and $1.2 \pm 0.3$[26]. In the GALPROP GDE model, the spatial gradient of CR densities in the very center region of the Galaxy is not well modeled because of its limited spatial resolution. Furthermore, the molecular hydrogen component incorporated in GALPROP is traced with the CO emission, which may be strongly contaminated by the foreground and background

emission in the GC. The CS radio map[29] is believed to be a better tracer of the molecular component[24,26]. Therefore we instead use the CS map in the CMZ region to construct the neutral pion decay $\gamma$-ray template. We take a power-law function $r^{-\alpha}$ to parameterize the radial distribution of the CR density in the CMZ (ignoring the contribution from the CR sea component) and multiply its line-of-sight integral with the CS emission template to model the spatial distribution of the hard component (referred to as the $CS \times r^{-\alpha}$ template hereafter; see methods, subsection The hard CR component in the CMZ). We scan $\alpha$ from 0.0 to 2.0 and obtained the difference of logarithmic likelihood values, defined as $\Delta\chi^2 = 2\ln\mathcal{L}_{max} - 2\ln\mathcal{L}_{\alpha,max}$, where $\mathcal{L}_{max}$ is the maximum likelihood of the fitting with varying $\alpha$ and $\mathcal{L}_{\alpha,max}$ is the maximum likelihood of the fitting with fixed $\alpha$. The $\Delta\chi^2$ value as a function of $\alpha$ is shown in Fig. 3, where a smaller value means a better fitting. We find that the Fermi-LAT data favors a non-zero value of $\alpha$ at a confidence level of 5.4σ. The inferred $\alpha = 1.35^{+0.06}_{-0.09}$ is consistent with those derived in the VHE band[24,26]. To visualize the radial dependence of the CR density and to investigate the dependence on the mass estimation with different observations, we derive the CR density in the CMZ region with the CS map divided into 3 segments, as done previously for the Planck map. Following ref. [29] we estimate the mass for each segment with the corresponding CS flux. As shown in panel (b) of Fig. 2, the results agree well with those incorporating the Planck map.

We derive the spectral energy distributions (SED) of the hard component in the CMZ region, using the $CS \times r^{-1.35}$ template, and report the results in Fig. 4. The spectral index of the $\gamma$-rays in the energy range from 8 to 500 GeV is $2.50 \pm 0.08$, which is roughly consistent with that measured in the VHE band ($2.32 \pm 0.05_{stat} \pm 0.10_{syst}$ and $2.28 \pm 0.03_{stat} \pm 0.20_{syst}$ by H.E.S.S.[24,25]). As a test of systematics, we also fit the data with the $CS \times r^0$ template and the GALPROP $H_2$ template to investigate the dependence of the CMZ CR spectrum on the spatial templates, and always get a hard spectrum. The fitting qualities of these two templates, however, are significantly poorer than that of the $CS \times r^{-1.35}$ template. For the $CS \times r^0$ (GALPROP $H_2$) template we have $-2\ln(\mathcal{L}) = 105826.6$ (105831.4), while for the $CS \times r^{-1.35}$ template we have $-2\ln(\mathcal{L}) = 105797.2$ (see also Fig. 3 for the differences of the log-likelihood values for different density profile slope $\alpha$). The $\gamma$-ray flux between 8 and 500 GeV in the inner 150 pc ($\pm1°$) region is $(5.50 \pm 0.38) \times 10^{-5}$ MeV cm$^{-2}$ s$^{-1}$. Taking the mass of dense molecular clouds as

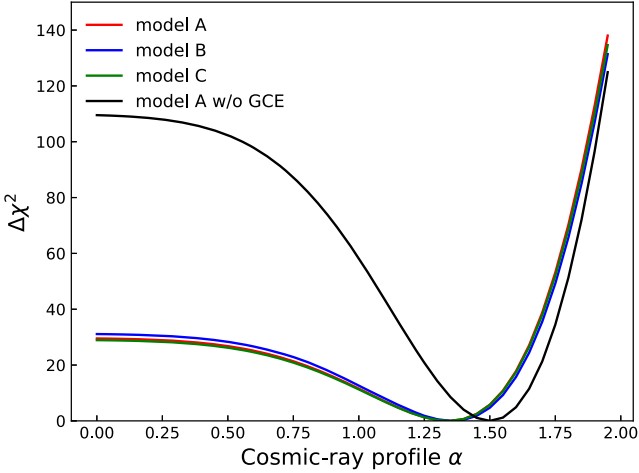

**Fig. 3 $\Delta\chi^2$ as a function of the power-law index of the CR density profile, $\alpha$.** The smaller the $\Delta\chi^2$, the better the fitting. Results, corresponding to the GDE models A, B, and C from ref. [33], with different assumptions on the source distributions, the diffusion coefficients, and the re-acceleration parameters of CRs (see Table 2 of ref. [33] for more details about different GDE models A, B, and C), for the off-CMZ region, are shown in red, blue, and green solid lines, respectively. The black solid line is for the fitting using the GDE model A without the inclusion of the GCE. Source data are provided as a Source Data file.

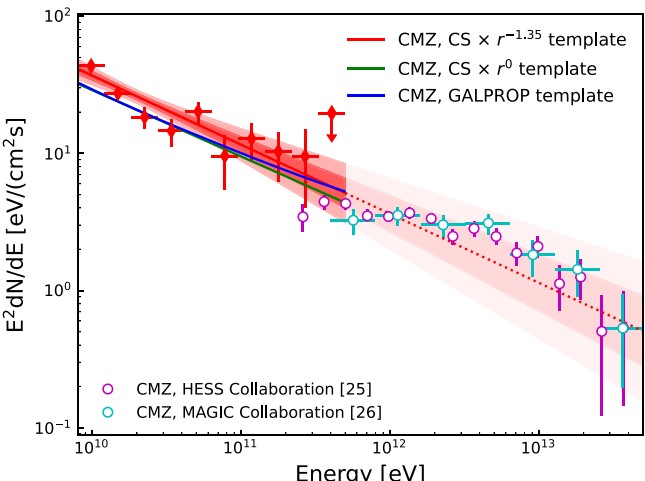

**Fig. 4 Spectral energy distribution of γ-ray emission contributed by the CMZ component.** Red filled dots show results obtained in this work, open circles in magenta show measurement by H.E.S.S.[25] and open circles in cyan show measurement by MAGIC[26]. The red solid line and associated dark and light bands show the best-fit power-law model and the 68 and 95% confidence ranges, respectively, for the $CS \times r^{-1.35}$ template. The error bars represent the $1\sigma$ statistical uncertainties and the upper limit is at the 95% confidence limit. Best-fit power-law models for the $CS \times r^0$ template and the GALPROP $H_2$ template are also shown in green and blue solid lines as a comparison. The lighter red band and the red dotted line above 500 GeV are extrapolations from the low energy fitting for the model with $CS \times r^{-1.35}$ template. Source data are provided as a Source Data file.

$3 \times 10^7 \, M_\odot$ [29,34] and the nuclear enhancement factor $\eta_N = 1.5$ for heavier nuclei (see "Methods", subsection The CR densities in the CMZ and off-CMZ regions), we have $\omega_{CR} = (4.5 \pm 0.3) \times 10^{-2}$ eV cm$^{-3}$ in the CMZ, which is close to the local CR density of $\sim 4.3 \times 10^{-2}$ eV cm$^{-3}$ in the same energy range[35]. The proton number density above 10 GeV, which is $(4.3 \pm 0.3) \times 10^{-12}$ cm$^{-3}$

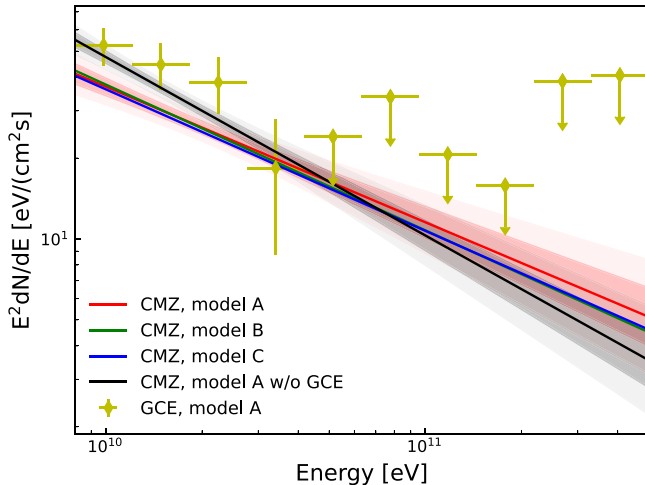

**Fig. 5 Energy fluxes of γ-ray emission from neutral pion decay associated with $H_2$ in the CMZ region.** To test the robustness of the results, different GDE models of the other emission components, such as the templates for emissions from the inverse Compton scattering or the neutral pion decay in the off-CMZ region, are used in the fitting. With the $CS \times r^{-1.35}$ template, the red solid line and associated dark and light bands are for results using GDE model A. The blue and green solid lines are for results using GDE model B and GDE model C, respectively. The black band and line are for the fitting using GDE model A but without incorporating the GCE. The SED of the GCE component is also shown in yellow points for reference. Dark and light regions show the 68 and 95% confidence ranges of the power-law fitting spectrum, respectively. The error bars represent the $1\sigma$ statistical uncertainties and the upper limits are at the 95% confidence level. Source data are provided as a Source Data file.

supposing the power-law spectrum holds for energies down to 10 GeV, is about 3 times smaller than the expectation of the CR sea in the GC region[36]. Note that the derived proton number density in the CMZ region in ref. [36] is also lower than that from the conventional model prediction, which supports our results.

## Discussion

The systematics from the choice of the GDE model could be important in affecting the results, as found in the previous studies[17,18,33,37]. Here we use several GDE models, namely models A, B, and C from ref. [33], with different assumptions on the source distributions, the diffusion coefficients, and the re-acceleration parameters of CRs (see Table 2 of ref. [33] for more details about different GDE models A, B, and C), to test the robustness of the results. Figure 5 shows the spectra of the hard component derived assuming the $CS \times r^{-1.35}$ template, for different GDE models of the other emission components. The results show good agreement with each other. Also, we scan $\alpha$ for these different GDE models, and find again very similar results (see Fig. 3).

The GCE could contribute to the diffuse γ-ray emission in the CMZ region. It has an intensity profile of $r^{-2\beta}$ with $\beta \sim 1.1$–$1.4$[18,38]. The spectrum of the GCE component is soft above a few GeV, as shown in Fig. 5 which is derived assuming a $r^{-2.56}$ profile (a line-of-sight projection is further applied when fitting the data). In the VHE band, the effect of GCE on the hard component should be negligible. At GeV energies, however, the flux of the GCE is higher than that of the hard component, and may affect the analysis significantly. Without addressing the GCE component in the analysis, we get a softer spectrum with an index of $2.64 \pm 0.06$ ($2.65 \pm 0.07$) for the CMZ region using a Planck template ($CS \times r^{-1.35}$ template), as shown in Fig. 1 (Fig. 5). This is

consistent with the inferred spectral index of CR protons in ref. [8], 2.80 ± 0.03, considering the fact that the energy-dependence of the inelastic interaction cross-section would result in a hardening of ~0.1 for γ-rays compared with CRs. We also compare the proton density at 10 GeV in the CMZ with previous work. In our analysis, it is $(1.00 \pm 0.05) \times 10^{-12}$ GeV$^{-1}$ cm$^{-3}$, which is close to the value for Sgr B complex in ref. [8]. Note that besides the CMZ region, the spectrum of the point source, Sagittarius A$^\star$, should also be influenced by the contamination from the GCE which is beyond the focus of this work. The impact of the GCE on the CR density distribution in the inner Galaxy region can be seen in panel (c) of Fig. 2. We can see that the CR density still declines quickly with increasing distance from the GC in the CMZ region, consistent with ref. [28], although in which the comparison between the CMZ and off-CMZ CR densities was not carried out. The background model without the GCE will increase the CR density in the innermost part of the GC region. For sky segments far away from the innermost GC, we can still observe a flux decrease from the off-CMZ region to the CMZ region, implying the barrier effect of the CR penetration. The impact of the GCE on the density profile of the hard CMZ component is shown in Fig. 3. The best-fit slope parameter α becomes steeper, ~1.5, if the GCE is not subtracted.

As found in the above analysis, the Fermi-LAT observations indicate that the contribution of the CR sea, which might be accelerated by sources distributed in the Galactic plane and diffuse across the whole Galaxy, should be suppressed in the CMZ. However, the γ-ray fluxes shown in Fig. 1 or the CR densities in Fig. 2 reflect the line-of-sight integrals of those quantities. To estimate the impacts due to the projection effect and also the uncertainties of the foreground and background modeling, we adopt the quasi-three-dimensional gas distributions used in the GALPROP package and apply randomly scaling factors to the gas annuli (except for the innermost annulus which is our target), which are in the face-on view of the Galactic plane and centered on the GC, based on the fitted distribution of the normalization factor, and re-fit the density contrasts of the off-CMZ region to the CMZ region. Please see "Method", subsection The foreground/background effect for the detailed description. We find that the CR density contrast is always larger than 2 for the segment with distance to the GC between 1.0 and 1.5°, consistent with that shown in Fig. 2. We conclude that the CMZ should indeed play a role to block CRs from entering into the very center region.

Several mechanisms can impede CR penetration into molecular clouds, such as the effect of magnetic field compression and the self-excited magnetohydrodynamics turbulence[39–41]. Taking an analogy of the solar modulation effect where low-energy CRs are blocked outside of the solar system by the magnetic field associated with solar winds, we briefly discuss the suppression of CR penetration in the CMZ. Employing the solar modulation model of ref. [42] as an example, we assume a spherical region of the CMZ with a radius of $R = 200$ pc and replace the solar wind velocity by the galactic wind velocity. The velocity of the galactic wind is not certain in the GC region[43–46], and we take it $V_0 \sim 500$ km s$^{-1}$ as a reference. We assume that the diffusion coefficient $D_0$ is constant in the CMZ, and neglects its energy-dependence. Note that this is different from ref. [42] in which the diffusion coefficient was assumed to be proportional to radius $r$. We have tested that assuming the same radius-dependence of the diffusion coefficient as ref. [42], the same scaling relation as obtained analytically can be reproduced. We solve numerically the transportation equation of particles

$$\frac{1}{r^2}\frac{\partial}{\partial r}\left(r^2 V_0 S - r^2 D_0 \frac{\partial S}{\partial r}\right) = \frac{q V_0 S}{r},\qquad(1)$$

where $S(r)$ is the spatial part of the particle distribution, which we separate from the energy part, $q \approx -1$ is related to the energy spectrum of particles. Adopting $D_0 = 3 \times 10^{28}$ cm$^2$ s$^{-1}$ (Note that $D_0$ depends on the chosen value of $V_0$ and is for CR particles with an energy of several hundreds of GeV), we can obtain a scaling relation of $\sim r^{0.8}$ for the CR density distribution for $50 < r/\text{pc} < 200$ (corresponding to $0.3°-1.3°$ angular range), which means the CR density decreases by a factor of ~3 from $r = 200$ pc to $r = 50$ pc. Therefore the CR fluxes for $r < 200$ pc can be efficiently suppressed. A hard CR component with a density profile of $\sim r^{-1.35}$ has been revealed by the Fermi-LAT data, which is consistent with that shown by the VHE observations. Taking the same framework of the CR transportation as discussed above, and assuming a steady point-like source at the GC, we obtain an approximate $r^{-1.5}$ profile of the CR density for $50 < r/\text{pc} < 200$, which is close to the observed value. Note that the pure diffusion predicts a $r^{-1}$ profile, and the convection results in a steeper profile. The Fermi-LAT data may imply a deviation from the $r^{-1}$ distribution, which could be a hint of the existence of convective winds. Considering that the diffusion coefficient should be energy-dependent, and at higher energies diffusion may be more important than convection, we may expect that the CR distribution in higher energy band is closer to $r^{-1}$ than in lower energies, which is just consistent with the observations. Alternatively, a time-dependent particle injection may also lead to a steeper radial profile. Detailed modeling is necessary, which is beyond the scope of the current work.

We note that the current analysis is probably limited by the projection effect of three-dimensional distributions of both the gas density and the CR density[26]. More precise three-dimensional gas models, particularly in the innermost region of the GC, may further improve our understanding of the CR origin as well as the transportation in the GC[34].

## Methods

**The Fermi-LAT data.** The Fermi-LAT data of version P8R3 and class SOURCE are used in this analysis. We select the data recorded from August 4, 2008 to February 1, 2020, in total 600 weeks. Since one of our goals is to investigate the spatial distribution of the CR density in the GC region, where crowded point sources may affect the morphology of the diffuse emission, data with good angular resolution are crucial in this analysis. In addition, the GCE which peaks around few GeV[17,18] may also significantly affect the low-energy analysis. We, therefore, select photons with energies higher than 8 GeV, which have a balance of a good angular resolution and sufficiently high statistics. To suppress the contamination from γ-ray generated by CR interactions in the upper atmosphere, photons collected at zenith angles larger than 90° are removed. Moreover, we adopt the specifications (DATA_QUAL > 0) && (LAT_CONFIG==1) to select good quality data. We bin the data, from 8 to 500 GeV, into 20 logarithmically distributed energy bins and 100 × 100 spatial bins with a pixel size of 0.1° centered at the GC. We employ the binned likelihood analysis method to analyze the data with the Fermitools version 1.2.1. The instrument response function (IRF) adopted is P8R3_SOURCE_V2.

**Point sources and the diffuse model components.** The source model XML file is generated using the user-contributed tool make4FGLxml.py based on the 4FGL catalog[47,48]. The spectrum of Sagittarius A$^\star$ is modeled as a power-law instead of the default log-parabola spectrum. Two additional point sources, 3FHL J1747.2-2822 and 3FHL J1748.6-2816, which are not in the 4FGL source catalog gll_psc_v23.xml but in the 3FHL source catalog[49] gll_psch_v12.xml, are also added in our analysis. Power-law spectra of the two 3FHL sources are assumed.

We use the GDE modeled by the GALPROP code in this work, which gives different components of the diffuse emission individually. The result of model A as introduced in ref. [33] is taken as our baseline GDE model. We also adopt their models B and C, which assume different CR source distributions and diffusion coefficient, for cross-check. For systematics studies about the template tracing the neutral pion decay γ-rays, besides the mentioned GALPROP generated model, we also employ the Planck dust opacity map from ref. [32] as an alternative, which traces the gas distributions without the uncertainty from the $X_{CO}$ factor of the conventional molecular gas tracer. The bremsstrahlung in our energy range is expected to be small and is thus neglected. The γ-ray emissions from the inverse Compton scattering of high-energy electrons off the optical, infrared, and the cosmic microwave background are taken as another template. The third diffuse template is the isotropic background. Power-law spectra of these three diffuse

templates are assumed, and both the normalizations and spectral indices are treated as free parameters in the likelihood fit. For the GCE component, we use a line-of-sight integration of $\rho^2(r)$, where $\rho(r) \propto r^{-1.28}$ as derived in ref. [18], to model its spatial distribution. Again we assume a power-law to model the spectrum of the GCE. As shown in ref. [18], the power-law spectrum is reasonable to describe the high-energy tail of the GCE.

To summarize, the model used in the likelihood fitting includes: point sources from the 4FGL and 3FHL catalogs, the inverse Compton scattering component of the GDE, the isotropic diffuse background, the GCE, and the neutral pion decay component of the GDE, with the last one being our target and various spatial templates are used during the analysis. To show in detail the $\gamma$-ray components used in our analysis, we list all the different components which are adopted in the fittings in a table supplied as Supplementary Table 1.

**The CR densities in the CMZ and off-CMZ regions**. To obtain the CR density in the CMZ region, the spatial template tracing the neutral pion decay is split into the CMZ region and off-CMZ region. The spectral indices and normalizations of the CMZ and off-CMZ regions are fitted independently.

To further reveal the CR density distribution without a pre-defined spatial template, we split the Planck template into segments as defined in Fig. 2. We model each segment with a power-law spectrum by fixing the spectral index to be 2.56 (2.64) in the CMZ (off-CMZ) region, i.e., the values derived from the previous fittings with CMZ and off-CMZ division. We fix all the parameters of the point sources outside the CMZ to their best-fitting values as well. The CR densities ($w_{CR}$ in eV/cm³) in those segments can thus be estimated from the normalizations (and hence luminosities) of the $\gamma$-ray emission as

$$w_{CR}(E_{CR}) \approx 0.018 \left(\frac{\eta_N}{1.5}\right)^{-1} \times \left(\frac{L_\gamma(E_\gamma)}{10^{34} \text{ erg/s}}\right) \left(\frac{M}{10^6 M_\odot}\right)^{-1}, \quad (2)$$

where $E_{CR} \approx 10 E_\gamma$ is the corresponding energy of CRs giving $\gamma$-ray energy of $E_\gamma$, $\eta_N$ accounts for the correction from nuclei heavier than protons and is taken as 1.5 in this work, $L_\gamma(E_\gamma)$ is the $\gamma$-ray luminosity, and $M$ is the total mass of the gas in the segment, which is estimated using the relation between the dust opacity and the column density[50].

**The hard CR component in the CMZ**. In order to compare with the VHE measurements consistently, we use the CS map multiplied the projected radial profile of the CR density to model the $\gamma$-ray emission from the CR interaction with the molecular gas in the CMZ region. For the neutral pion decay template generated by GALPROP, we mask the contribution from $H_2$ traced by CO in the CMZ to avoid double-counting it. The contributions from atomic and ionized hydrogen, which are expected to be sub-dominant, are kept as foreground in the GDE model. Note that the background CR sea may also contribute to the $\gamma$-ray emission in the CMZ even if most of it is blocked due to the barrier effect of the CMZ. This may add up to the hard CR component. We neglect it in the current treatment. Also, the hard CR component should extend to the off-CMZ region. Due to its fast decline, we do not consider it outside the CMZ region.

**The foreground/background effect**. To examine the effect due to the foreground and background, we use the outputs of GALPROP predicted templates calculated from the Galactocentric annuli of gas as an approximation of a three-dimensional modeling. We rebin the gas file into 6 annuli, which are in the face-on view of the Galactic plane and centered on the GC, with radii of [0, 1.5) kpc, [1,5, 3.5) kpc, [3.5, 5.5) kpc, [5.5, 8) kpc, [8, 10) kpc, and [10, 50) kpc, respectively. We take the first segment, from 0 to 1.5 kpc, as our target, and split it further into the CMZ region and off-CMZ region. In the fitting yielding Fig. 1 with the GALPROP template, the normalization and index for the off-CMZ component, which is the same for all annuli, are fitted to be $1.19 \pm 0.03$ and $2.63 \pm 0.04$, respectively. As an approximation, for the five annuli except the innermost one, we fix their spectral index to be 2.63, but randomly assign the normalization parameters according to a Gaussian distribution with a central value of 1.19 and standard deviation of 0.15 (5 times larger than that obtained in the fitting). All other components (three-point sources in the CMZ, the inverse Compton scattering GDE, the isotropic diffuse emission, the GCE, and the CMZ and off-CMZ regions of the innermost annulus) are left free to fit. With this setup, we expect that the projection effect from the region outside the innermost annulus will be reduced. We further divide the innermost annulus into segments as shown in panel (a) of Fig. 2 and fit the normalizations of each segment. We find that the CR density ratios of the off-CMZ to the CMZ segment for distance bin from 1.0° to 1.5° to the GC are mostly larger than 2, confirming the finding that the CR density in the CMZ is lower than that outside the CMZ. However, we should also note that CO emission, instead of CS emission, is used to trace the distribution of molecular hydrogen near the GC in the GALPROP, and this may lead to unaccounted for systematics.

## Data availability
The Fermi-LAT observation data analyzed/used in this work are publicly available (https://heasarc.gsfc.nasa.gov/FTP/fermi/data/lat/weekly/photon/). The Fermi-LAT 4FGL catalog could be found at https://fermi.gsfc.nasa.gov/ssc/data/access/lat/

10yr_catalog/and the Fermi-LAT 3FHL catalog could be found at https://fermi.gsfc.nasa.gov/ssc/data/access/lat/3FHL/. The Planck dust opacity map is available at https://irsa.ipac.caltech.edu/data/Planck/release_2/all-sky-maps/maps/component-maps/foregrounds/COM_CompMap_Dust-GNILC-Model-Opacity_2048_R2.01.fits. The CS map is available at https://www.nro.nao.ac.jp/~nro45mrt/html/data/tsuboi/GC.CS10.10.FITS.tar.gz. The model maps of DGE models A, B, and C could be found at http://www-glast.stanford.edu/pub_data/845/. Source data are provided with this paper. The data that support the findings of this study are available from the corresponding author upon reasonable request. Source data are provided with this paper.

## Code availability
The Fermitools used in this work are publicly available (https://fermi.gsfc.nasa.gov/ssc/data/analysis/software/). GALPROP is available at http://galprop.stanford.edu/. The user-contributed tool make4FGLxml.py for generating source model XML file is available at https://fermi.gsfc.nasa.gov/ssc/data/analysis/user/make4FGLxml.py.

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

## Acknowledgements
We acknowledge the use of the Fermi-LAT data provided by the Fermi Science Support Center. We thank Zhaoqiang Shen and Bei Zhou for useful discussion. This work is supported by the National Natural Science Foundation of China (Nos. 11921003, U1738205, U1738210), the Key Research Program of the Chinese Academy of Sciences (No. XDPB15), Chinese Academy of Sciences, and the Program for Innovative Talents and Entrepreneur in Jiangsu.

## Author contributions
X.H. carried out the data analysis. X.H., Q.Y. and Y.-Z.F. interpreted the data and prepared the manuscript.

## Competing interests
The authors declare no competing interests.
