## [Peer Review File · Nature Communications]

REVIEWER COMMENTS

Reviewer #1 (Remarks to the Author):

The topic of the paper represents a great interest, that is why I agreed to review this article. But I should admit that it was quite difficult for me to understand the results and conclusions. I passed several times through the text, but still, I am confused by most of the statements. Therefore, before formulating my assessment, I need clarifications of several questions. I cannot move on without having answers to the questions below.

1. Do I understand correctly that the authors claim the presence of three components of GeV gamma-radiation linked to the Central Molecular Zone (CMZ): (1) so-called Galactic Center GeV excess (GCE), (2) the radiation induced by the CR sea, and (3) gamma-rays produced by particles accelerated in the Galactic Center?

2. If yes, do I understand correctly that the first (GCE) component was not analyzed/investigated in this work; namely, its presence was postulated and the spectral and radial characteristics were taken from previous studies of the other authors?

3. Do I understand correctly that assuming that the sum of the 2nd and 3rd components of gamma-rays is derived using different templates produced by interactions of protons and nuclei with the ambient gas, as well as by subtraction of the postulated 1st component? If so, then am I right that the authors have derived the spectral and radial distributions of cosmic rays contributed by the CR sea and by the particles accelerated in the central part of CMZ - hereafter CRI and CCII components, respectively.

4. Am I right that you claim that

(i) for the CRI component, you have found (1) the energy spectrum similar to the spectrum of the locally measured CR spectrum, and (2) dramatic suppression of their density towards the GC.

(ii) for CCII you found a harder energy spectrum and an increase of the cosmic ray density towards the GC, i.e. just opposite radial dependence compared to the profile of CRI.

5. How the parameters characterizing the spectral and radial distributions of these two components have been separated?

I have many other comments and questions, but I would like to be sure that I understand correctly your major points. Also, please try to make the manuscript more readable. Presently it contains many confusing sentences. The confusions start already in the abstract. For example, what do you mean when claiming

“This work reports the identification of GeV-TeV cosmic rays in the central molecular zone with the γ -ray observations of the Fermi-LAT” (lines 14-15”). Do you claim a discovery of a “new GeV-TeV particle component* as it is indicated in the title (line 4)? Furthermore, in the next sentence “The spectrum and spatial gradient of the GeV-TeV source component are consistent with that measured by the Imaging Atmospheric Cherenkov Telescopes” (lines 15-16) you argue that the Fermi Data indicate a smooth continuation of the gamma-ray spectrum of CMZ measured by the Cherenkov telescopes in the TeV range towards low energies. Although this is a reasonable and most likely a correct statement, it certainly cannot be called a discovery since that has been already indicated in some previous studies as well (see e.g. Malyshev et al 2015, A&A, 582, A11, Chernyakova et al. 2011, ApJ, 726, id. 60)).

The next sentence “The inferred cosmic-ray energy density is, however, substantially lower than the so-called cosmic-ray sea component” (lines 16-17) is even more confusing. The density of which component of cosmic rays is lower than the density of the sea? I can only guess what do you mean, but I am sure.

Anyway, I think the manuscript needs a major revision, with a clear explanation of the procedures used in the data analysis, strict definitions and clear statements. Only after that, I will be able to provide more specific comments and conclude whether the paper could be considered for publication or not.

Reviewer #2 (Remarks to the Author):

The author's argue that there is a screening mechanism stopping cosmic rays entering the central molecular zone. I think their analysis is sound and their conclusions sufficiently interesting to warrant publication in nature communications.

Reviewer #3 (Remarks to the Author):

Dear editor,

The authors present an interesting study of the properties of the diffuse cosmic-ray (CR) population inside the central molecular zone. This is achieved by using gamma-ray events pointing back to that region, from a decade long exposure with Fermi-LAT. Using a multi-component likelihood fit that includes point-like sources as well as specially extended features they try to single out the contribution that originates from neutral pion decay after CR interactions with dense molecular gas in that region. This allows them to infer properties of the underlying CR population.

The two key results from that study are the existence of a radial CR density profile that decays with r to the power of -1.35 and peaks at the dynamic center of our Galaxy and an overall decreased flux density of CRs in the energy range between 80 GeV and 5 TeV when comparing the central molecular zone (CMZ) to the surrounding regions.

While the first of these findings is not an entirely new feature per se, it is the first time that it has been confirmed for this (lower) energy regime and nicely matches with measurements at higher energies. The second main result on the other hand seems not to have been discovered/pointed out yet and might have some quite interesting implications for the understanding of the production mechanisms and propagation of CRs in the center of our Galaxy. This may be due to the fact that the energy regime probed by the authors is at the higher end of the Fermi energy range where this component significantly contributes to the overall gamma-ray flux. The authors suggest that a screening mechanism must be at work in that region that efficiently prevents CRs from the surrounding CR sea to enter the CMZ. A scenario similar to the solar wind creating a bubble that shields CRs at lower energies is suggested.

I have the feeling that the results presented in this manuscript have some quite interesting implications for the understanding of the propagation of CRs and the configuration of magnetic fields within the CMZ, but also that the topic will have to be re-addressed at some point using more data and especially an improved 3d modeling of the gas distribution (something also already hinted at by the authors).

I have the following comments regarding the content:

Section 2:

The general methodology applied throughout this manuscript seems to be sound (multi-component, binned spectral and spatial joint likelihood fit), unfortunately the details of the model are not always very clear to the reader. It would be very helpful if the authors would provide a table listing all the different components that went into the fit (point-sources can be combined into groups). This is specifically important as the authors fit different scenarios/combinations and it should be made easier for the reader to get an overview. It is not clear which of the diffuse model components are fitted separately in the CMZ and off-CMZ regions (line 89). Only the neutral pion decay component or also the other components from the GALPROP model(s)?

In some places (l. 91/92, l. 95/96, l. 141 are most prominent) the authors make references to other measurements/publications, giving a citation, but then instead of citing also the actual values they compare to, only give relative statements. This makes it very hard for the reader to verify the statements as he has no other option than searching for that value in the original publication. Citing those numbers would be highly appreciated.

In lines 137/138 the authors state that they also used the GALPROP template and the CS map without radial scaling for fitting the SED of the CMZ component, yielding similar results but a poorer quality of the fit. This statement is not supported by any numbers or graphics like residual map(s). Any such information/material would be helpful in order to convey to the reader a sense for the degree of worsening of the fit if changing the spatial model.

In line 139 the gamma-ray flux of the CMZ component is converted to an energy density of CRs by taking the mass of the (total?) mass of the molecular clouds from (Tsuboi et al. 1999). Wouldn't it make sense to repeat the analysis that is displayed in Figure 2, using the CS map instead of the Planck dust opacity map? The differences between both scenarios could be used as an indicator for systematic uncertainties from the uncertainties in the knowledge of the true distribution of target material.

One last point regarding this section that I would like the authors to comment on is how Figure 8 panel (b) of reference (Acero et al. 2016) relates to the work presented here. I have two main questions here:

- Is this where the CR sea expectation (l. 141) refers to?

- How do the authors regard the first data point close to 0 kpc in relation to their work? Is this already a hint for the screening of CRs at the GC that was previously overlooked or taken for a systematic effect, but supports their claim? I am not aware of any publication that pointed this out explicitly however.

Section 3:

3.1 (Figure 5): Why is this test not done also for the CS template?

The authors study the influence of introducing the “2 GeV” Galactic Center Emission (GCE) component (or not) into their fit, but not that much attention is given to the influence of projection effects and fore-/backgrounds. To my understanding only a singular model component is used for modeling CR interactions, inside the CMZ, where also other in the fore-/backgrounds on the line of sight might play a role. It could be interesting to use some way (maybe an interpolation of the “out of CMZ” outside template) of fitting the CMZ on top of a fore-/background that is also determined by the emission from surrounding regions. After all this could further strengthen the claim of the authors that the CR density inside the CMZ is significantly lower than outside.

While the scenario suggested in 3.3 in order to explain the flux deficit of GeV – TeV CRs in the CMZ sounds reasonable to me, it should be pointed out that my expertise does not allow me to make a definitive statement on that.

General/minor:

The plots in the paper do generally not make a very high quality impression in their current form, in terms of formatting and choice of colors. Figure 4 and especially Figure 5 have too many different error bands/butterflies over-plotted at too high opacity for the reader to be able to disentangle.

Figure 4: I don't like that the authors draw the error regions/butterflies of their SED fits way beyond the actual energy range that went into the fit. Constraining those to the actual energy range of their data and plotting an extrapolation with for example dotted lines or lower opacity would not only highlight better the energy range of the data but also make the plot more clear in my opinion.

The language of the manuscript definitely needs some polishing, especially in the first half. As a non-native speaker I am probably not the perfect person for a thorough language correction, but am just naming a few instances:

- l. 18 ... of a high energy ...
- l. 24 ... possibly experiencing reacceleration, convection ...
- l. 29 ... of fresh CRs ... of such a fresh ...
- l. 31 ... black hole, Sagittarius A*, and ...
- l. 40 ... might have been more active ...
- l. 43 ... for a possible counterpart ...
- l. 53 ... emission signals ...
- l. 62 ... and a large enough statistic. ... (?)
- l. 65 ... select good quality data ...
- l. 88 for studying what?
- l. 155 ... comparable (slightly higher) to ... (?)

REVIEWER COMMENTS

Reviewer #1 (Remarks to the Author):

The topic of the paper represents a great interest, that is why I agreed to review this article. But I should admit that it was quite difficult for me to understand the results and conclusions. I passed several times through the text, but still, I am confused by most of the statements. Therefore, before formulating my assessment, I need clarifications of several questions. I cannot move on without having answers to the questions below.

1. Do I understand correctly that the authors claim the presence of three components of GeV gamma-radiation linked to the Central Molecular Zone (CMZ): (1) so-called Galactic Center GeV excess (GCE), (2) the radiation induced by the CR sea, and (3) gamma-rays produced by particles accelerated in the Galactic Center?

Reply: Yes, you do. In the analysis we consider such three components. To be more specific, each of these components consists of two parts, inside and outside the CMZ. For the GCE that might have a dark matter or astrophysical origin, we assume a unified power-law spatial profile as determined by a large-scale analysis of it. The 2nd and 3rd components should overlap with each other and are hard to be isolated. However, since the spatial distribution of the emission in the CMZ shows a decrease from the GC to outside and the density of CR in the CMZ is lower than that outside the CMZ, we expect that the emission inside the CMZ mainly comes from the 3rd component and outside the CMZ the emission mainly comes from the 2nd component.

In the analysis, we also include point sources in the 4FGL and 3FHL catalogs. To be more readable, we have added some clarification of the emission components in the third paragraph of page 2 as well as in the Methods (see page 8). Following the third referee's suggestion, we also provide a table as supplementary materials to show explicitly the model components used in our analysis.

2. If yes, do I understand correctly that the first (GCE) component was not analyzed/ investigated in this work; namely, its presence was postulated and the spectral and radial characteristics were taken from previous studies of the other authors?

Reply: No, we have included the GCE component in this analysis. Its spatial distribution, $r^{-2.56}$, which was obtained via a larger scale analysis (Calore et al. 2015), is fixed. This is because relaxing the spatial distribution of the GCE in the small region of the current work would result in strong degeneracy with other components. The spectrum of the GCE component (including both index and normalization) is left free to be fitted. The influence of GCE is demonstrated in Fig.2, Fig.3, and Fig.5.

3. Do I understand correctly that assuming that the sum of the 2nd and 3rd components of gamma-rays is derived using different templates produced by interactions of protons and nuclei with the ambient gas, as well as by subtraction of the postulated 1st component? If so, then am I right that the authors have derived the spectral and radial distributions of cosmic rays contributed by the CR sea and by the particles accelerated in the central part of CMZ - hereafter CRI and CCII components, respectively.

Reply: Yes, you are right.

The 2nd and 3rd components in the CMZ region are very difficult to be isolated, and we treat them as a single component at first. The 1st component (the GCE) was subtracted via a joint fitting in the analysis. We single out the CMZ region, and get the spectral distribution in the CMZ. Then we study the radial distributions of CRs in the central region of the GC, in segments with different distance from the GC, as can be seen in Fig. 2. The spatial distribution of the CR densities is different from the expectation that the smoothly distributed CR sea would be overlaid with a new component of fresh CRs. This may indicate that the contribution of the CR sea (i.e., CRI by the referee) should be suppressed in the CMZ, and the emission inside the CMZ mainly comes from a new component (i.e., CRII by the referee), which is associated with the VHE CR component.

4. Am I right that you claim that

(I) for the CRI component, you have found (1) the energy spectrum similar to the spectrum of the locally measured CR spectrum, and (2) dramatic suppression of their density towards the GC.

Reply: Yes, you are right.

For the CRI component (equivalently the CR sea) in the off-CMZ region, the index of the spectrum is 2.64 for gamma-rays, corresponding to about 2.74 for CRs, and is similar to the locally measured CR spectrum. The CR density keeps roughly a constant in the off-CMZ region. However, the density gets suppressed significantly in the CMZ region.

(ii) for CRII you found a harder energy spectrum and an increase of the cosmic ray density towards the GC, i.e. just opposite radial dependence compared to the profile of CRI.

Reply: For the CRII component (equivalently the new CR component accelerated in the CMZ), its spectrum is 2.50 ± 0.08 (i.e., harder than the CR sea component), and the spatial profile is about $r^{-1.35}$. We do not get the radial profile of CRI in the CMZ region. But from a comparison of the fluxes in the off-CMZ and CMZ regions, we find that the CRI component should be suppressed and be subdominant in the CMZ.

5. How the parameters characterizing the spectral and radial distributions of these two components have been separated?

Reply: In the CMZ region, the radial distribution of the CR density shows a power-law form of nearly $r^{-1.35}$, consistent with that obtained in the VHE band. Also the SED of the CMZ emission agree well with that measured by HESS and MAGIC. We thus think that the CRII component is mainly attributed to the new component in the CMZ and is the low-energy counterpart of the VHE source. In the off-CMZ region, the background sea component should dominate due to the quick decrease of the CRI component. The energy spectrum is close to that expected from the CR sea, and the spatial distribution is flat. These properties are consistent with the scenario that CRI is dominated by the CR sea. Furthermore, the CMZ may also simultaneously block CRI component from escaping into the off-CMZ region.

We have added related discussion in the end of "CRs with low density in the CMZ" part on page 3, and in the Methods part on page 8.

I have many other comments and questions, but I would like to be sure that I understand correctly your major points. Also, please try to make the manuscript more readable. Presently it contains many confusing sentences. The confusions start already in the abstract. For example, what do you mean when claiming

“This work reports the identification of GeV-TeV cosmic rays in the central molecular zone with the γ -ray observations of the Fermi-LAT” (lines 14-15”). Do you claim a discovery of a “new GeV-TeV particle component* as it is indicated in the title (line 4)? Furthermore, in the next sentence “The spectrum and spatial gradient of the GeV-TeV source component are consistent with that measured by the Imaging Atmospheric Cherenkov Telescopes” (lines 15-16) you argue that the Fermi Data indicate a smooth continuation of the gamma-ray spectrum of CMZ measured by the Cherenkov telescopes in the TeV range towards low energies. Although this is a reasonable and most likely a correct statement, it certainly cannot be called a discovery since that has been already indicated in some previous studies as well (see e.g. Malyshev et al 2015, A&A, 582, A11, Chernyakova et al. 2011, ApJ, 726, id. 60)).

The next sentence “The inferred cosmic-ray energy density is, however, substantially lower than the so-called cosmic-ray sea component” (lines 16-17) is even more confusing. The density of which component of cosmic rays is lower than the density of the sea? I can only guess what do you mean, but I am sure.

Anyway, I think the manuscript needs a major revision, with a clear explanation of the procedures used in the data analysis, strict definitions and clear statements. Only after that, I will be able to provide more specific comments and conclude whether the paper could be considered for publication or not.

Reply: Thanks for pointing out this. We have made substantial efforts to improve the presentation of the manuscript, including the structure of the paper (following the editor's request). Hopefully this new version clarifies the confusion parts and is more readable now.

This work reports two results as indicated in the title: the identification of a counterpart of the VHE accelerator in GeV-TeV energy range using the Fermi-LAT data, and the finding of a barrier effect of CRs from penetrating into the CMZ region. These two results were not really discovered or realized before. For

example, Malyshev et al. (2015) and Chernyakova et al. (2011) focused on the point source possibly associated with Sgr A*. Acreo et al. (2016) showed the decrease of CR intensities in the GC region compared with larger scale results. However, as a work concentrating on building the model of galactic interstellar emission, these authors did not look into the CR density variations in the very central region. Particularly the change of CR density around the boundary of the CMZ as revealed in our work was not shown in Acero et al. (2016). Gaggero et al. (2017) carried out the small-scale analysis. However, they focused on the region inside the CMZ, and did not reveal the CR density changes inside/outside the CMZ. Also in all these work, the possible contamination from the GCE are ignored. Therefore we think that the findings in our work are novel and deserve publication.

For "The inferred cosmic-ray energy density is, however, substantially lower than the so-called cosmic-ray sea component", we mean the cosmic-ray energy density in the CMZ region, which is dominated by the contribution from the CR II component, is lower than the cosmic-ray energy density derived from the off-CMZ region, the so-called cosmic-ray sea component.

Thanks to your and the other referees' very helpful suggestions/comments, the current version has got significantly improved. We are looking forward to your favorable recommendation.

#####

Reviewer #2 (Remarks to the Author):

The author's argue that there is a screening mechanism stopping cosmic rays entering the central molecular zone. I think their analysis is sound and their conclusions sufficiently interesting to warrant publication in nature communications.

Reply: Thank you very much for the kind support!

#####

Reviewer #3 (Remarks to the Author):

Dear editor,

The authors present an interesting study of the properties of the diffuse cosmic-ray (CR) population inside the central molecular zone. This is achieved by using gamma-ray events pointing back to that region, from a decade long exposure with Fermi-LAT. Using a multi-component likelihood fit that includes point-like sources as well as specially extended features they try to single out the contribution that originates from neutral pion decay after CR interactions with dense molecular gas in that region. This allows them to infer properties of the underlying CR population.

The two key results from that study are the existence of a radial CR density profile that decays with r to the power of -1.35 and peaks at the dynamic center of our Galaxy and an overall decreased flux density of CRs in the energy range between 80 GeV and 5 TeV when comparing the central molecular zone (CMZ) to the surrounding regions.

While the first of these findings is not an entirely new feature per se, it is the first time that it has been confirmed for this (lower) energy regime and nicely matches with measurements at higher energies. The second main result on the other hand seems not to have been discovered/pointed out yet and might have some quite interesting implications for the understanding of the production mechanisms and propagation of CRs in the center of our Galaxy. This may be due to the fact that the energy regime probed by the authors is at the higher end of the Fermi energy range where this component significantly contributes to the overall gamma-ray flux. The authors suggest that a screening mechanism must be at work in that region that efficiently prevents CRs from the surrounding CR sea to enter the CMZ. A scenario similar to the solar wind creating a bubble that shields CRs at lower energies is suggested.

I have the feeling that the results presented in this manuscript have some quite interesting implications for the understanding of the propagation of CRs and the configuration of magnetic fields within the CMZ, but also that the topic will have to be re-addressed at some point using more data and especially an improved 3d modeling of the gas distribution (something also already hinted at by the authors).

Reply: We thank the referee for the positive comments on this work. We also appreciate you for the following very detailed/helpful suggestions/comments.

I have the following comments regarding the content:

Section 2:

The general methodology applied throughout this manuscript seems to be sound (multi-component, binned spectral and spatial joint likelihood fit), unfortunately the details of the model are not always very clear to the reader. It would be very helpful if the authors would provide a table listing all the different components that went into the fit (point-sources can be combined into groups). This is specifically important as the authors fit different scenarios/combinations and it should be made easier for the reader to get an overview. It is not clear which of the diffuse model components are fitted separately in the CMZ and off-CMZ regions (line 89). Only the neutral pion decay component or also the other components from the GALPROP model(s)?

Reply: The components used in the fitting model are indeed complicated since there are many components in the vicinity of the GC. Basically we have point sources from the 4FGL and 3FHL catalogs, the GCE component, the isotropic diffuse background, the GDE component from the GALPROP model, and the new CR component in the CMZ. For the GDE component, we neglect the bremsstrahlung component and consider only the inverse Compton scattering component and the pion-decay component. For the new CR component in the CMZ, only the pion-decay emission is assumed. In the line 89, only the pion-decay component are fitted separately for the CMZ and off-CMZ regions. The inverse Compton scattering component is treated commonly for the CMZ and off-CMZ regions.

We have added some clarification of the emission components in the Methods of page 8. We also provide a table as supplementary materials to show explicitly the model components used in our analysis.

In some places (l. 91/92, l. 95/96, l. 141 are most prominent) the authors make references to other measurements/publications, giving a citation, but then instead of citing also the actual values they compare to, only give relative statements. This makes it very hard for the reader to verify the statements as he has no other option than searching for that value in the original publication. Citing those numbers would be highly appreciated.

Reply: We have included these values in the revised version.

In lines 137/138 the authors state that they also used the GALPROP template and the CS map without radial scaling for fitting the SED of the CMZ component, yielding similar results but a poorer quality of the fit. This statement is not supported by any numbers or graphics like residual map(s). Any such information/material would be helpful in order to convey to the reader a sense for the degree of worsening of the fit if changing the spatial model.

Reply: The logarithmic likelihood values of the fitting with different templates have been added. For the CS $\times r^{\{0\}}$ (GALPROP HS_2) template we have $-\ln(\{\mathcal{L}\})=105826.6$ (105831.4), while for the CS $\times r^{\{-1.35\}}$ template we have $-\ln(\{\mathcal{L}\})=105797.2$. The differences of the log-likelihood values for different density profile slope α can also be seen in Fig. 3. We have added these values in the revised manuscript.

In line 139 the gamma-ray flux of the CMZ component is converted to an energy density of CRs by taking the mass of the (total?) mass of the molecular clouds from (Tsuboi et al. 1999). Wouldn't it make sense to repeat the analysis that is displayed in Figure 2, using the CS map instead of the Planck dust opacity map? The differences between both scenarios could be used as an indicator for systematic uncertainties from the uncertainties in the knowledge of the true distribution of target material.

Reply: The mass used for the conversion is the estimated molecular mass in the inner 150 pc region of the CMZ. The analysis using the CS template in the CMZ region is done following the suggestion. Since the CS template covers only the CMZ region, we use the GALPROP template for the off-CMZ region. The results are in agreement with that using the Planck dust map. We have added the results in Fig. 2.

One last point regarding this section that I would like the authors to comment on is how Figure 8 panel (b) of reference (Acero et al. 2016) relates to the work presented here. I have two main questions here:

- Is this where the CR sea expectation (l. 141) refers to?
- How do the authors regard the first data point close to 0 kpc in relation to their work? Is this already a hint for the screening of CRs at the GC that was previously overlooked or taken for a systematic effect, but supports their claim? I am not aware of any publication that pointed this out explicitly however.

Reply: The solid line in the panel (b) of Figure 8 of Acero et al. (2016) is the CR sea expectation. So the results of Acero et al. (2016) did show that the CR intensity in the GC region is smaller than the prediction of the conventional model of the CR sea. However, these authors paid attention to an all-sky analysis of the GDE, without looking into the CR density variations in the very central region. Particularly the change of CR density around the boundary of the CMZ as revealed in our work was not shown in Acero et al. (2016). Thus the screening effect of the CRs was overlooked, though there might be hint to show this. We think that the results of Acero et al. (2016) actually support our claim in this paper. We have added some discussion about this in the revision (page 4).

Section 3:

3.1 (Figure 5): Why is this test not done also for the CS template?

Reply: In fact, the test in Figure 5 is done for the CS $\propto r^{-1.35}$ template of the CMZ component and for different GDE models of the other emission components (the inverse Compton scattering component, the HI and HII GDE component, and the H2 component in the off-CMZ region). We have clarified this in the caption and also in the text.

The authors study the influence of introducing the “2 GeV” Galactic Center Emission (GCE) component (or not) into their fit, but not that much attention is given to the influence of projection effects and fore-/backgrounds. To my understanding only a singular model component is used for modeling CR interactions, inside the CMZ, where also other in the fore-/backgrounds on the line of sight might play a role. It could be interesting to use some way (maybe an interpolation of the “out of CMZ”

outside template) of fitting the CMZ on top of a fore-/background that is also determined by the emission from surrounding regions. After all this could further strengthen the claim of the authors that the CR density inside the CMZ is significantly lower than outside.

Reply: We thank the referee for pointing out this important issue. We agree with the referee that the projection effect of the line-of-sight emission may affect somehow the quantitative results of the current analysis. A rigorous treatment of the projection effect may need a precise modeling of the three-dimensional gas distribution of the Milky Way, which is, however, difficult at the current stage.

We have done a test to approximately account for such an effect, based on the quasi-three-dimensional gas distributions as used in the GALPROP tool. We rebin the gas file into 6 annuli with radii of [0, 1.5) kpc, [1.5, 3.5) kpc, [3.5, 5.5) kpc, [5.5, 8) kpc, [8, 10) kpc, and [10, 50] kpc, respectively. We take the first segment, from 0 to 1.5 kpc, as our target, and split it further into the CMZ region and off-CMZ region. In the fitting yielding Fig. 1 with the GALPROP template, the normalization and index for the off-CMZ component, which is the same for all annuli, are fitted to be 1.19 ± 0.03 and 2.63 ± 0.04 , respectively. As an approximation, for the five annuli except the innermost one, we fix their spectral index to be 2.63 , but randomly assign the normalization parameters according to a Gaussian distribution with central value of 1.19 and standard deviation of 0.15 (5 times larger than that obtained in the fitting). All other components (three point sources in the CMZ, the inverse Compton scattering GDE, the isotropic diffuse emission, the GCE, and the CMZ and off-CMZ regions of the innermost annulus) are left free to fit. With this setup, we expect that the projection effect from the region outside the innermost annulus will be reduced. We further divide the innermost annulus into segments as that shown in panel (a) of Fig. 2, and fit the normalizations of each segment. We find that the CR density ratios of the off-CMZ to the CMZ segment for distance bin from 1.0 to 1.5 degrees to the GC are mostly larger than 2, confirming the finding that the CR density in the CMZ is lower than that outside the CMZ.

We have added the above discussion in the Discussion part on page 5 and Methods part on pages 8-9.

While the scenario suggested in 3.3 in order to explain the flux deficit of GeV – TeV CRs in the CMZ sounds reasonable to me, it should be pointed out that my expertise does not allow me to make a definitive statement on that.

General/minor:

The plots in the paper do generally not make a very high quality impression in their current form, in terms of formatting and choice of colors. Figure 4 and especially Figure 5 have too many different error bands/butterflies over-plotted at too high opacity for the reader to be able to disentangle.

Figure 4: I don't like that the authors draw the error regions/butterflies of their SED fits way beyond the actual energy range that went into the fit. Constraining those to the actual energy range of their data and plotting an extrapolation with for example dotted lines or lower opacity would not only highlight better the energy range of the data but also make the plot more clear in my opinion.

Reply: We have re-plotted these figures to make them clearer.

The language of the manuscript definitely needs some polishing, especially in the first half. As a non-native speaker I am probably not the perfect person for a thorough language correction, but am just naming a few instances:

- 1. 18 ... of a high energy ...
- 1. 24 ... possibly experiencing reacceleration, convection ...
- 1. 29 ... of fresh CRs ... of such a fresh ...
- 1. 31 ... black hole, Sagittarius A*, and ...
- 1. 40 ... might have been more active ...
- 1. 43 ... for a possible counterpart ...
- 1. 53 ... emission signals ...
- 1. 62 ... and a large enough statistic. ... (?)
- 1. 65 ... select good quality data ...
- 1. 88 for studying what?
- 1. 155 ... comparable (slightly higher) to ... (?)

Reply: Thanks for point out these issues. We have tried to improve the presentation in the revised version.

REVIEWER COMMENTS

Reviewer #1 (Remarks to the Author):

I want to thank the authors for the detailed answers to my questions and appreciate the corresponding changes in the revised manuscript. Now the statements of the authors sound straightforward and more convincing. I am generally satisfied with the answers, therefore I am inclined to recommend the paper for publication. But before doing so, I have some additional comments:

1. The Fermi-LAT data match the energy and radial energy distributions of radiation at TeV energies. Thus it is natural to conclude that the GeV and TeV components and correspondingly the parent CR protons are of the same origin. In this regard, I think it is misleading when the authors call it “New CR component”. “New” in which sense? The reader could take it as an addition to the component responsible for the TeV emission. In fact, most likely, we deal with the same component. If the authors agree with this statement, they should avoid calling it a “new component”.

2. One of the main conclusions of this paper is the suppression of the CR sea in the CMZ zone is in clear contrast to ref.[26] (Gaggero et al., 2017). The radial dependence found in this paper is in conflict with the claim of [26]. I think this should be clearly indicated; otherwise, the reader could be confused.

3. The analysis of gamma-ray data from CMZ is a rather complex procedure using different background templates, assumptions, approximations. This introduces uncertainties which, hopefully, should not dramatically change the conclusions of this paper. On the other hand, the information about the CR protons (the absolute flux and the spectral shape) derived from gamma-rays of the part of CMZ, namely from the Sgr B complex, perfectly agrees with the CR sea: see

<https://www.aanda.org/articles/aa/pdf/2015/08/aa25233-14.pdf>

<https://journals.aps.org/prd/pdf/10.1103/PhysRevD.101.083018>

The authors should comment and explain the reason for the discrepancy.

Reviewer #3 (Remarks to the Author):

Dear editor,

The authors have implemented most of my previous suggestions and also answered the questions I had to my satisfaction. Overall, the restructured manuscript now makes a more clear impression to me. But, as also pointed out by the first referee regarding the first version, it still is comparably hard to read.

It has to be acknowledged that the type of analysis that is presented here is quite complex by nature and that any results from such a study will always bear a certain degree of model dependence. This should incentivize the authors to very clearly state the motivations behind all the decisions made in terms of the choice of the model (components). It should also be taken as a reason for very thoroughly and in a detailed way describing the applied procedure. Having said this, I still would like to recommend to the authors to spell out things more clearly in this regard wherever they see a chance, although being aware of the complexity of the subject.

Before going into concrete comments/suggestions I would like to make clear that regardless of the details of the analysis and the way the results are presented, which one might argue could still be improved a bit, the authors, in sum, provide compelling evidence for an apparent lack of the CR-sea contribution in the GC. This alone should be considered a remarkable result and serves as reasonable motivation for the speculated-on barrier mechanism.

Comments/Suggestions:

I. 37/38: I do not fully agree with this statement. The spatial distribution from such a component is not expected to be flat. This component is simply ignored by some analyses, although the generally expected profile should be $1+r^{-\alpha}$ (sea + diffusion from source). Maybe you can rephrase this sentence.

I. 54: there is a barrier in the CMZ → there is a barrier surrounding the CMZ? Or say “The CMZ acts as a barrier to CRs entering from outside”?

I. 58: CRs with low density in the CMZ. → Reduced (GeV?) CR density within the CMZ ?

l. 61 .. and caption of FIG. 1: In neither of the two locations the difference between the red and green error bands is properly explained. It should be stated (in the caption and in the text already around line 60) that two different spatial models have been used for the fit. I know that the Planck template is mentioned later, but the context of the values given in l. 68 is not clear at this point.

FIG. 1: I would still reduce the alpha value on those error bands for a clearer display.

l. 74 .. : Maybe it would be worth mentioning here that FIG. 2b shows the results from two different fits (with and without the GCE component)

FIG. 2: I would like the authors to at least consider to present the data points of the right panel that result from a fit without the GCE component in a separate panel. The crowded plot makes it hard to immediately take away the crucial bit of information from this figure.

Caption of FIG. 2: orangeand → orange and

l. 99: using the relation between the total CS fluxes and the mass ← can you put a citation here?

l. 108: (see also Fig. 3) ← I think that I understand the point you want to make here, but putting it a bit more explicitly would definitely not harm.

Caption of FIG. 3: Can you mention here what exactly is the difference between models A,B and C or at least put a reference to literature about that? It is a reoccurring scheme in this manuscript that as a reader one stumbles over plots corresponding to different assumptions/models in the Figures, without proper explanation in the caption or in the preceding/referring text, only to find such an explanation in a later part of the text. Maybe you can find a way to improve this by either mentioning the explanation earlier in the text, or in the caption or putting some sort of reference for pointing the reader to the right place.

l. 141: The gas annuli that are mentioned here are are annuli in a face-on view of the Galaxy? This is not made very clear in this paragraph.

FIG. 5: Wouldn't it make sense to also compare the CMZ component w/o GCE to the CR-sea component w/o GCE? Dashed lines like in FIG. 1 for example.

Also here I suggest to use a lower alpha value for the error band.

l. 231: Also here the meaning of the annuli does not become 100% clear, although I assume that you mean annuli in the face-on view of the Galactic plane and centered on the GC.

l. 244: unaware → unaccounted for ?

REVIEWER COMMENTS

Reviewer #1 (Remarks to the Author):

I want to thank the authors for the detailed answers to my questions and appreciate the corresponding changes in the revised manuscript. Now the statements of the authors sound straightforward and more convincing. I am generally satisfied with the answers, therefore I am inclined to recommend the paper for publication. But before doing so, I have some additional comments:

Reply: Thanks for the support!

1. The Fermi-LAT data match the energy and radial energy distributions of radiation at TeV energies. Thus it is natural to conclude that the GeV and TeV components and correspondingly the parent CR protons are of the same origin. In this regard, I think it is misleading when the authors call it “New CR component”. “New” in which sense? The reader could take it as an addition to the component responsible for the TeV emission. In fact, most likely, we deal with the same component. If the authors agree with this statement, they should avoid calling it a “new component”.

Reply: We agree that this GeV component is the low energy part of the TeV component and they share the same origin from the cosmic-ray interaction. We have removed "new" in the title. In the main text, we have changed the wording "new" to "hard".

2. One of the main conclusions of this paper is the suppression of the CR sea in the CMZ zone is in clear contrast to ref.[26] (Gaggero et al., 2017). The radial dependence found in this paper is in conflict with the claim of [26]. I think this should be clearly indicated; otherwise, the reader could be confused.

Reply: In fact, in the CMZ region, both studies of ref.[26] and our work revealed a declining radial profile. But for the region outside the CMZ, ref.[26] did not derive the density of the CR sea component with a consistent treatment as those for the CMZ region, and thus did not reach the conclusion of CR intensity suppression in the CMZ. Furthermore, ref.[26] did not scrutinize contributions from different diffuse components in the CMZ region, but assume that all

emission, after subtracting point source contribution using Fermi's catalog parameters, is from the neutral pion decay. As shown in our analysis, in the CMZ region, the GCE would have a sizable gamma-ray flux, and may affect the analysis significantly. The different data analysis processes and background assumptions may lead to the difference of these two studies. We have added a discussion in the revised manuscript (page 5).

3. The analysis of gamma-ray data from CMZ is a rather complex procedure using different background templates, assumptions, approximations. This introduces uncertainties which, hopefully, should not dramatically change the conclusions of this paper. On the other hand, the information about the CR protons (the absolute flux and the spectral shape) derived from gamma-rays of the part of CMZ, namely from the Sgr B complex, perfectly agrees with the CR sea: see

<https://www.aanda.org/articles/aa/pdf/2015/08/aa25233-14.pdf>;
<https://journals.aps.org/prd/pdf/10.1103/PhysRevD.101.083018>;

The authors should comment and explain the reason for the discrepancy.

Reply: In the analysis we tested different background models, as shown in figure 3 and figure 5, and found that the conclusion is insensitive on these models. The well matched spectra for the central component from several GeV to tens of TeV also indicate that our treatment for the background is reasonable.

As for the comparison with the references mentioned by the referee, we expect again that the difference might be mainly due to the GCE component. As we have tested, the obtained gamma-ray spectrum is softer if the GCE is excluded, such as 2.65 ± 0.07 for the $CS \times r^{-1.35}$ template and 2.64 ± 0.06 for the Planck template. In <https://journals.aps.org/prd/pdf/10.1103/PhysRevD.101.083018> the spectral index of CR protons was found to be about 2.80 ± 0.03 for the Planck template, the corresponding gamma-ray index would be ~ 0.1 harder, which is thus consistent with what we found above. We also compare the proton density at 10 GeV in the CMZ. In our analysis, it is $(1.00 \pm 0.05) \times 10^{-12} \text{ GeV}^{-1} \text{ cm}^{-3}$ without incorporating the GCE, which is also close to the value in Table III of <https://journals.aps.org/prd/pdf/10.1103/PhysRevD.101.083018>.

101.083018 for Sgr B complex. We have added some discussion about such comparisons in the revised manuscript (page 5).

#####

Reviewer #3 (Remarks to the Author):

Dear editor,

The authors have implemented most of my previous suggestions and also answered the questions I had to my satisfaction. Overall, the restructured manuscript now makes a more clear impression to me. But, as also pointed out by the first referee regarding the first version, it still is comparably hard to read.

It has to be acknowledged that the type of analysis that is presented here is quite complex by nature and that any results from such a study will always bear a certain degree of model dependence. This should incentivize the authors to very clearly state the motivations behind all the decisions made in terms of the choice of the model (components). It should also be taken as a reason for very thoroughly and in a detailed way describing the applied procedure. Having said this, I still would like to recommend to the authors to spell out things more clearly in this regard wherever they see a chance, although being aware of the complexity of the subject.

Before going into concrete comments/suggestions I would like to make clear that regardless of the details of the analysis and the way the results are presented, which one might argue could still be improved a bit, the authors, in sum, provide compelling evidence for an apparent lack of the CR-sea contribution in the GC. This alone should be considered a remarkable result and serves as reasonable motivation for the speculated-on barrier mechanism.

Reply: Thank you again for the supportive suggestions/comments. We have tried to include more information, especially in captions of figures, in the manuscript to clarify the choice of models. Hopefully such efforts would have made the manuscript more understandable.

Comments/Suggestions:

l. 37/38: I do not fully agree with this statement. The spatial distribution from such a component is not expected to be flat. This component is simply ignored by some analyses, although the generally expected profile should be $1+r^{-\alpha}$ (sea + diffusion from source). Maybe you can rephrase this sentence.

Reply: We agree that in the GC region the smoothly distributed CR sea would be overlaid with an additional component of fresh CRs, which would lead to the generally expected ($1+r^{-\alpha}$) profile (see the first sentence of the second paragraph). The sentence has been rephrased in our revision.

l. 54: there is a barrier in the CMZ → there is a barrier surrounding the CMZ? Or say “The CMZ acts as a barrier to CRs entering from outside”?

Reply: Corrected as suggested.

l. 58: CRs with low density in the CMZ. → Reduced (GeV?) CR density within the CMZ ?

Reply: Corrected as suggested.

l. 61 .. and caption of FIG. 1: In neither of the two locations the difference between the red and green error bands is properly explained. It should be stated (in the caption and in the text already around line 60) that two different spatial models have been used for the fit. I know that the Planck template is mentioned later, but the context of the values given in l. 68 is not clear at this point.

Reply: We have added related descriptions in both the caption and the text.

FIG. 1: I would still reduce the alpha value on those error bands for a clearer display.

Reply: The plot has been updated.

l. 74 .. : Maybe it would be worth mentioning here that FIG. 2b shows the results from two different fits (with and without the GCE component)

FIG. 2: I would like the authors to at least consider to present the data points of the right panel that result from a fit without the GCE component in a separate panel. The crowded plot makes it hard to immediately take away the crucial bit of information from this figure.

Caption of FIG. 2: orangeand → orange and

Reply: We have added panel (c) for the results without the GCE component as suggested. The caption is also updated accordingly. The typo is corrected.

l. 99: using the relation between the total CS fluxes and the mass ← can you put a citation here?

Reply: Added.

l. 108: (see also Fig. 3) ← I think that I understand the point you want to make here, but putting it a bit more explicitly would definitely not harm.

Reply: We have added some words to make it clear.

Caption of FIG. 3: Can you mention here what exactly is the difference between models A,B and C or at least put a reference to literature about that? It is a reoccurring scheme in this manuscript that as a reader one stumbles over plots corresponding to different assumptions/models in the Figures, without proper explanation in the caption or in the preceding/referring text, only to find such an explanation in a later part of the text. Maybe you can find a way to improve this by either mentioning the explanation earlier in the text, or in the caption or putting some sort of reference for pointing the reader to the right place.

Reply: We have clarified this in the caption.

l. 141: The gas annuli that are mentioned here are are annuli in a face-on view of the Galaxy? This is not made very clear in this paragraph.

Reply: Yes, they are. We have clarified it.

FIG. 5: Wouldn't it make sense to also compare the CMZ component w/o GCE to the CR-sea component w/o GCE? Dashed lines like in FIG. 1 for example. Also here I suggest to use a lower alpha value for the error band.

Reply: It would be difficult to use the method in Fig. 1 to do the comparison here, since in Fig. 5 we use the CS map which is not a sky survey but only covers the CMZ region. So we made this comparison in Fig. 1 using the Planck template and this would also help us to compare with the result derived in Sgr B complex as suggested by Reviewer #1.

We have also updated Fig. 5.

l. 231: Also here the meaning of the annuli does not become 100% clear, although I assume that you mean annuli in the face-on view of the Galactic plane and centered on the GC.

Reply: Yes. We have clarified it in the text.

l. 244: unaware → unaccounted for ?

Reply: Corrected.

REVIEWERS' COMMENTS

Reviewer #1 (Remarks to the Author):

I am satisfied with the changes and the response to my comments.

I recommend publishing the paper provided that Reviewer #3 also is satisfied with the response to his/her comments.

REVIEWERS' COMMENTS

Reviewer #1 (Remarks to the Author):

I am satisfied with the changes and the response to my comments.

I recommend publishing the paper provided that Reviewer #3 also is satisfied with the response to his/her comments.

Reply: Thanks for the support!

Reviewer #3 (Remarks to the Author):

Reply: Thanks for the support!